# A Systematic Review of Genetic Polymorphisms Associated with Bipolar Disorder Comorbid to Substance Abuse

**DOI:** 10.3390/genes13081303

**Published:** 2022-07-22

**Authors:** Adriano de Marco, Gabriele Scozia, Lucia Manfredi, David Conversi

**Affiliations:** 1Department of Psychology, Università degli Studi di Roma ‘La Sapienza’, 00185 Rome, Italy; demarco.1598754@studenti.uniroma1.it (A.d.M.); gabriele.scozia@uniroma1.it (G.S.); lucia.manfredi@uniroma1.it (L.M.); 2PhD Program in Behavioral Neuroscience, Università degli Studi di Roma ‘La Sapienza’, 00185 Rome, Italy

**Keywords:** bipolar disorder, substance use disorder, polymorphisms

## Abstract

It is currently unknown which genetic polymorphisms are involved in substance use disorder (SUD) comorbid with bipolar disorder (BD). The research on polymorphisms in BD comorbid with SUD (BD + SUD) is summarized in this systematic review. We looked for case-control studies that genetically compared adults and adolescents with BD and SUD, healthy controls, and BD without SUD. PRISMA was used to create our protocol, which is PROSPERO-registered (identification: CRD4221270818). The following bibliographic databases were searched indefinitely until December 2021 to identify potentially relevant articles: PubMed, PsycINFO, Scopus, and Web of Science. This systematic review, after the qualitative analysis of the study selection, included 17 eligible articles. In the selected studies, 66 polymorphisms in 29 genes were investigated. The present work delivers a group of potentially valuable genetic polymorphisms associated with BD + SUD: rs11600996 (*ARNTL*), rs228642/rs228682/rs2640909 (*PER3*), PONQ192R (*PON1*), rs945032 (*BDKRB2*), rs1131339 (*NR4A3*), and rs6971 (*TSPO*). It is important to note that none of those findings have been confirmed by two or more studies; thus, we believe that all the polymorphisms identified in this review require additional evidence to be confirmed.

## 1. Introduction

Extreme mood swings with alternating depressive episodes and manic, hypomanic, or mixed periods are the hallmarks of bipolar disorder (BD). BD’s prevalence is estimated at around 1–1.5%, equally divided into the two sexes [1], and its manifestations vary in phenomenology and severity. Three primary types of BD are recognized by the Diagnostic and Statistical Manual of Mental Disorders (DSM-5; American Psychiatric Association, 2013): bipolar disorder I, bipolar disorder II, and cyclothymic disorder. BD is a concrete public health problem, representing the sixth cause of disability in people between 15 and 44 years old. Women are more likely to have rapid cycles and mixed states or comorbidity patterns that differ from males, including lifetime eating disorders [2]. BD can also occur in association with other diseases. Indeed, 60–65% of patients with BD depend on legal or illegal psychoactive substances, especially alcohol, tobacco, cannabis, cocaine, heroin, and hallucinogens [3]. In addition, the intake of those substances tends to bring out comorbid mental disorders in predisposed subjects, anticipating their onset, worsening their manifestations, and greatly complicating the pharmacological and psychosocial management of patients in therapy, thus becoming a fundamental index of severity [4]. The DSM-5 defines substance use disorders (SUDs) with at least two of 11 criteria in 12 months, with different levels of disorder severity at a higher presence of criteria (2–3 = mild; 4–5 = moderate; ≥6 = severe). The criteria for SUDs can apply to a variety of substances (such as alcohol, nicotine, cannabis, opioids, and cocaine), and they include increased use, ineffective attempts to cut back or stop, continued use despite unfavorable social, psychological, and physical effects, persistent craving, tolerance development, and withdrawal symptoms [5].

The genetic underpinnings involved in the pathophysiology of BD comorbid to SUD are currently unclear. In recent years, due to GWAS technologies, the knowledge about disease risk genes has been improved. In the case of BD, GWAS studies found several genetic markers, and many genes and polymorphisms overlap with markers associated with schizophrenia or major depression. This could suggest that BD can be regarded as a point on a spectrum of risk, ranging from major depression to schizophrenia. Therefore, the genetics and inherited risk for BD represent a challenge in understanding this psychiatric disease. In the literature, there is an agreement in supporting the idea that at least three genes are associated with bipolar disorder. One of the first genes to be involved in BD by GWAS was *ANK3*, located on chromosome 10q21.2 [6,7,8,9,10]. Several studies suggest a significant association between BD and SNPs near *ANK3*, which directly affects the expression of *ANK3*. This gene encodes a protein involved in axonal myelination, ankyrin B, with expression in multiple tissues, especially in the brain. The second gene associated with BP is *CACNA1C*, located on chromosome 12p13. The gene encodes an L-type voltage-gated ion channel with recognized functions in neural development and synaptic communication, which has been implicated in numerous investigations of BP, schizophrenia, and major depression. Without a clear syndromic resemblance to BD, heterozygous disruption of the gene in mice changes several behaviors assumed to reflect mood [11,12,13,14,15,16,17,18,19,20,21]. The last third gene associated with BD and schizophrenia is *TRANK1*, on chromosome 3p22. *TRANK1* encodes a protein highly expressed in multiple tissues, especially in the brain, and could be involved in maintaining the blood–brain barrier. It is reported that valproic acid mood stabilizer therapy increases the expression of *TRANK1*, and cells carrying the risk allele show reduced expression of the gene and its protein. It has also been suggested, by recent transcriptomic studies, that the dysregulation of another gene in the 3p22 locus, *DCLK3*, may contribute to risk for both BD and schizophrenia. This gene is predicted to have different functions: in particular it has been involved in the protein kinase activity and in the peptidyl-serine phosphorylation. Other functions regard the upstream/downstream regulation of protein localization to nucleus and, finally, it has been suggested to be active in cytoplasm and nucleus [22,23,24,25,26,27,28,29].

The methodology of large GWAS samples has facilitated recent advances in SUD genetics. It is confirmed that SUDs are highly polygenic disorders, with many variants across the genome conferring risk. SUDs may involve the use of multiple substances, and for this reason, the relevant information is reported from various branches of substance abuse studies. First, it is pertinent noting that genome-wide association studies have reported more reliable results when examining genetic associations with licit substances, such as alcohol and nicotine. In contrast, studies that examined the genetic association with illicit substances, such as cocaine and cannabis, reported less reliable and replicable data due to the small sample sizes.

Nonetheless, here, we report interesting and relevant data about the potential genetics of different substances. Important insights come from the literature that investigated the genetics of alcohol use disorder (AUD): larger sample sizes have helped to detect critical loci in the *ADH1B* gene (e.g., rs1229984 and rs2066702) associated with AUD and with various measures of alcohol use and consumption [30,31,32,33,34,35,36,37,38,39,40,41]. Similarly, loci in the *ALDH2* (i.e., rs671) have been associated with alcohol dependence and alcohol-related traits (i.e., maximum drinks, flushing response) in East Asian and Thai populations [42,43,44] and for alcohol drinking status in East Asian people [33]. A large multi-ethnic genome-wide association study has replicated the association of this *ALDH2* polymorphism with alcohol drinker status and drinks/week in an East Asian population, whereas a genome-wide significant association has been reported in non-Hispanic whites between genetic variants of *ADH1B* (rs1229984) and alcohol use phenotypes. [45]. Further evidence associated *DRD2* (Dopaminergic Receptor D2) polymorphisms and AUD and problematic alcohol use [34,35,36]; Glucokinase Regulator gene (*GCKR* gene), Klotho Beta (*KLB*) gene variants, and *SLC39A8* variants are also associated with AUD and alcohol use problems, as well as consumption [33,36,37,39,40,46]. Another dependence widely investigated is the nicotine dependence (ND), which is consistently linked to cholinergic nicotinic receptor genes. For example, it is reported a gene-wide significant association between ND and genetic variations within the *CHRNA5*-*CHRNA3*-*CHRNB4* locus on chromosome 15 [47]. Further evidence for a top variant association in *CHRNA5* on chromosome 15 (rs16969968), and a significantly associated variant (rs151176846) in *CHRNA4* on chromosome 20 (*CHRNA4*) comes from the Nicotine Dependence Genomics Consortium (iNDiGO) [48,49]. In the field of illicit drugs, the small sample sizes implied fewer replicable genome-wide discoveries for cannabis use disorder (CanUD). Nonetheless, the most extensive GWAS study of CanUD identified two different gene-wide significant loci: *FOXP2* on chromosome 7 and *CHRNA2* on chromosome 8. The first has a role in synaptic plasticity and speech and language development, and the polymorphisms associated with CanUD (rs7783012) have been previously associated with externalizing behaviors. The second gene encodes the α-2 subunit of the neuronal nicotinic acetylcholine receptor, also linked to tobacco smoking and schizophrenia [50]. Another crucial substance abused in SUDs is cocaine. A gene-wide significant association with cocaine use disorder (CocUD) is rs2629540, located in the *FAM53B* gene [51]. Huggett and Stallings [52,53], moreover, identified four significant genes: *C1QL2*, *STK38*, and *KCTD20* in European Americans, and *NDUFB9* in African Americans. Another gene is determined by a meta-analysis, *HIST1H2BD*, also finding genetic correlations with schizophrenia, ADHD, major depression, and risk-taking [54,55].

In conclusion, although there is a comprehensive framework about the genetics of these psychiatric disorders, there is still a lack of genetic understanding of these pathologies in comorbidity. To this aim, this systematic review highlights and summarizes the research on polymorphisms in bipolar disorder in comorbidity with substance use disorders.

We looked for case-control studies with a genetic comparison between subjects with BD and SUD, healthy controls (CTRs), and BD without SUD.

## 2. Materials and Methods

### 2.1. Protocol and Registration

The protocol for the current study was written in accordance with Preferred Reporting Items for Systematic Reviews and Meta-Analyses: The PRISMA Declaration (PRISMA) [56]. The protocol is also included in the PROSPERO International Prospective Register of Systematic Reviews (CRD42021270818).

### 2.2. Eligibility Criteria

In the review were included: (1) studies authored in English; (2) empirical studies on polymorphisms potentially related to BD comorbid to SUD; (3) studies with both a Control Group and a BD group without SUD. Excluded the following studies: (1) animal studies; (2) studies on non-BD patients with other disorders of affective processes, such as schizophrenia and major depression; (3) reviews and meta-analyses.

### 2.3. Information Sources

The following bibliographic databases have been used to identify potentially relevant documents with no time constraint till December 2021: Scopus, PubMed, Web of Science, and PsycINFO. The findings were exported to Mendeley.

### 2.4. Search

The following keywords was used in all databases: (bipolar disorder) AND ((substance use disorder) OR (dependence) OR (substance abuse) OR (addiction)) AND ((polymorphisms) OR (genetic) OR (genes)). Filters were: humans; English; adult: >18 years, adolescence.

### 2.5. Study Selection

The study selection has been administered with a precise protocol. First, the evaluators had to decide whether the relevant articles’ titles and abstracts were admissible based on the inclusion criteria. As a result, two reviewers independently screened all publication titles and abstracts to ascertain whether the papers fulfilled the inclusion criteria. The publications on which both reviewers agreed advanced to the next step. If the two reviewers could not really agree on the validity of the article, a third reviewer was brought in to help come to a decision. The next step was to select manuscripts that investigated at least one polymorphism in BD comorbid SUD. The motivations led to the exclusion of all other studies:Polymorphisms in other disorders (right topic and wrong population), e.g., polymorphisms related to schizophrenia;No analyses on polymorphisms, but focus on BD (wrong topic and right population), e.g., treatment of BD;Neither analyses on polymorphisms nor discussion of BD (wrong topic and wrong population), e.g., treatment of schizophrenia.

In the final step, after the reviewers had identified the articles that passed the previous step, they read the full text in order to determine which study should be included in the qualitative analysis according to the selection criteria.

### 2.6. Data Collection Process

Two reviewers independently charted data from each eligible article, implementing them on Google Sheets. Any discrepancies were settled by discussion between the two reviewers or additional analysis by a third reviewer. All results were available to the entire review team.

### 2.7. Data Items

We abstracted data on article characteristics (author(s) and year of publication), study populations (groups, number of subjects, sex, age, diagnostic criteria of BD), identified polymorphisms, methods and study design, and main findings.

### 2.8. Risk of Biases in Individual Studies

We assessed each study using the Newcastle–Ottawa quality rating scale (NOS) [57]. The NOS is a classic assessment tool that evaluates the various studies’ selection, comparability, and exposure. This scale ranges from 0 to 9; an investigation is considered high quality if it achieves a score greater than 7. A study gains one star each time a reviewer thinks a specific question is answered clearly (for example, “Is the Case Definition Adequate?”). A study can only receive one star for each numbered item in the Selection and Exposure categories. For comparability, a maximum of two stars can be awarded. Two reviewers assigned NOS scores independently to all eligible studies, and if any disagreement was present, it was resolved by a third reviewer.

## 3. Results

### 3.1. Study Selection

We identified 2380 publications in the initial research on the different scientific databases. After removing duplicates, we analyzed 1445 articles in the first screening stage. among these, 1365 were eliminated due to the following factors: review, *n* = 244, incorrect topic but right population, *n* = 512, and wrong topic and incorrect population, *n* = 490. The second stage of the screening process included a comprehensive text analysis of the remaining 80 articles. Among these, 63 were removed for the incorrect population: non-BD group, *n* = 4; absence of BD group without SUD, *n* = 59. Due to their eligibility, the remaining 17 articles were analyzed qualitatively as part of the systematic review. A flowchart of the search procedure is shown in Figure 1.

### 3.2. Study Characteristics

The current systematic review focused on English-language articles published between 2000 and 2018. Most patients in all studies are female. Case subjects (BD group, BD + SUD group) ranged in age from 16 to 58 years, while control subjects ranged in age from 16 to 52 years. DSM-IV criteria were primarily used to assess BD. Case-control studies are always used. Within the chosen studies, 66 polymorphisms in 29 genes were examined. The features of the 17 articles included in the systematic review are summarized in Table 1.

### 3.3. Risk of Bias within Studies

The mean of the NOS score of the included studies was 7.35 (range 6–8), indicating that most studies were considered high quality. Potential study biases resulted mainly from baseline characteristics between controls and cases and inappropriate selection of control groups (see Appendix A for further information).

### 3.4. Results of Individual Studies

The 17 selected publications identified different kinds of genes. The following sections reported the results of the studies for different categories: serotonergic genes, dopaminergic genes, Circadian rhythm regulation genes, and other and unspecified genes.

#### 3.4.1. Serotonergic Genes

In total, 2 of the 17 selected articles investigate four polymorphisms of two serotonergic genes in BD comorbid to SUD: *HTR1B* and *5HTR2C* [66,68]. No significant relationship between these two genes and BD + SUD has been reported.

Huang and colleagues did not suggest a direct association with BP + SUD. The authors hypothesized that alcoholism, substance abuse, and suicidal behavior could be associated with the 861C allele. They analyzed the G861C polymorphism of the *HTR1B* gene. Substance abuse disorder and major depression appear to be associated with the *h5-HTR1B* G861C locus in the patient population. Other psychopathologies, such as bipolar disorder, schizophrenia, alcoholism, and suicide attempts, were not found to be associated with this polymorphism.

To test the hypothesis of an involvement of *5HTR2C* in BD, specifically in BD comorbid with SUD, Mazza and colleagues investigated three *5HTR2C* genetic variations in high-linkage disequilibrium (LD) in a sample of BD patients with or without a diagnosis of comorbid SUD and healthy controls. Statistical analysis showed that the G allele of the rs6318 polymorphism of the *5HTR2C* gene was significantly more frequent in BD patients (although the association is in the opposite direction of that previously reported: other positive studies found an increase in the frequency of the rarer C allele). The analysis of *5HTR2C* markers in BD + SUD patients did not yield significant results.

#### 3.4.2. Dopaminergic Genes

Two studies examined the role of five polymorphisms of four dopaminergic genes in BD comorbid SUD: *DRD1*, *DRD2*, *DRD3*, and *DRD4* [63,74]. No significant relationship between these two genes and BD + SUD has been reported.

Gorwood and colleagues have tested the role of the gene encoding the D2 dopamine receptor (TaqI A1 allele) in the potentially shared vulnerability to alcohol dependence and bipolar disorder. To detect the specific role of the D2 gene, the authors examined four samples to measure the interaction between bipolar disorder and alcoholism: alcoholic bipolar patients, alcohol-dependent patients, bipolar patients, and controls. According to the potential confounding effect of gender in the study (dependence is more frequent in males and mood disorders affect females preferentially), the control sample was limited to males. They did not find any significant evidence for a relationship between the A1 allele of the D2 dopamine receptor gene and the specific association between bipolar disorder and alcohol dependence.

Szczepankiewicz and colleagues investigated the possible relationship between polymorphisms in the four dopamine receptor genes (−48 A/G in *DRD1*, −141 C ins/del in *DRD2*, 9 Ser/Gly in *DRD3*, and −521 C/T in *DRD4*) and the comorbidity of alcohol abuse in bipolar patients. The authors hypothesized a common genetic background for these two disorders for the dual diagnosis of bipolar disorder and alcohol abuse. No association of any polymorphisms in the dopamine genes was found comparing bipolar patients in the comorbid alcohol abuse group with the healthy control group. The authors suggested that the analyzed polymorphisms may not be involved in the shared genetic vulnerability to bipolar disorder and alcohol abuse.

#### 3.4.3. Circadian Rhythm’s Regulation Genes

One selected study examined the potential role of Circadian rhythm’s regulation genes. In particular, Benach and colleagues [58] investigated the possible link among polymorphisms of *CLOCK*, *ARNTL*, *TIMELESS*, and *PER3* genes (complex analysis of single polymorphisms and haplotypes–SNP interactions) and the comorbidity of alcohol abuse (AAD) in bipolar disorder patients. An association of two polymorphisms, one from the *ARNTL* gene (rs11600996) and the other from the *PER3* gene (rs228642), with an increased risk of BD/AAD in a group of male patients has been found. They have also found that two other polymorphisms of the *PER3* gene, rs228682 and rs2640909, were associated with AAD and a family history of affective disorders.

#### 3.4.4. Other Genes

Ten studies investigated the role of fifty-one polymorphisms of seventeen other genes in BD comorbid to SUD: *PON1*, *ALDH2*, *ADH1B*, *NGFB*, *GHRH*, *PREB*, *GROK5*, *BDKRB2*, *CHRNB3*, *CHRNA5*, *CHRNA3*, *IL1b*, *NOS1AP*, *TRPM2*, *NR4A1*, *NR4A2*, *NR4A3*, *TSPO*, and *TRACR1*.

Bortolasci and colleagues [59] conducted two studies: the first to examine *PON1* status (*PON1* activity and functional PONQ192R polymorphism based on a 2-substrate assay) in patients with major depression and bipolar disorder with nicotine dependence. Major depression was accompanied by lowered *PON1* activity. *PON1* activity was decreased by smoking and a diagnosis by genotype interaction. The authors suggested that reduced plasma *PON1* activity may be a trait marker of major depression and that PONQ192R gene–environment (smoking) interactions differentially predict depression and bipolar disorder odds. In the second study, Bortolasci and colleagues [60] examined whether total radical trapping antioxidant potential (TRAP) levels are associated with *PON1* status, smoking, mood disorders, interactions between the *PON1* Q192R genotypes and smoking, ethnicity, marital status, body mass index (BMI), and gender. The authors reported various results. They observed that the TRAP levels were significantly associated with higher plasma *PON1* activity, the RR available genotype, non-smoking by RR carriers, male gender, and a higher body mass index (BMI). TRAP levels were significantly lower in mood disorder patients than in controls. Still, this association was no longer significant after the authors considered the effects of the predictors above. A smoking X RR genotype interaction raises the risk in the subgroup with low TRAP levels, while male gender, the RR genotype, higher BMI, and *PON1* activity lower it. Plasma *PON1* activity, *PON1* Q192R functional genotypes, and specific interactions between this genotype and smoking contribute significantly to TRAP levels. Gender and BMI appear to affect TRAP levels as well.

Cui and colleagues [62] investigated 130 genes for their association to primary affective disorders (PAFDs) and substance dependence comorbid to affective disorders (CAFDs) by case-control association analysis using a candidate gene array. The authors focused on the gene encoding nerve growth factor-beta (*NGFB*). *NGFB* is of interest in this regard for several reasons. First, NGFB showed the strongest association among the 130 candidate genes in the initial data mining analysis. Second, the functions of the NGFB protein are related to the neurotrophic hypothesis of Affective Disorders development. Nerve growth factor (NGF), a member of the neurotrophin family, consists of three subunits (alpha, beta, and gamma), apparently with the beta unit (NGFB) being solely responsible for the biological activity of NGF. NGFB is linked to pathways (PI3K/Akt, Ras/Raf/ERK, and PLC/PKC) crucial for the development, patterning, and maintenance of the nervous system, and these pathways are targets of mood stabilizers. To find interesting relationships, the authors analyzed fourteen polymorphisms of the *NGFB* gene (rs1146611, rs6330, rs6328, rs6326a, rs2268793, rs910330, rs2239622, rs2856813, rs6678788, rs4529705, rs6537860, rs10776799, rs4332358, and rs3811014). The authors reported a strong significant association of the *NGFB* gene with PAFDs in women but not with SUDs comorbid with Affective Disorders.

Gratacòs and colleagues [64] analyzed 338 candidate genes and 1029 SNPs. They have found results for five polymorphisms (rs1016164, rs2289359, rs10414815, rs945032, and rs8016905) of four genes: *GHRH* (growth hormone-releasing hormone, a gene that encodes a member of the glucagon family of proteins), *PREB* (prolactin regulatory element, a gene that encodes a protein that specifically binds to a Pit1-binding element of the prolactin promoter), *GRIK5* (Ionotropic kainate glutamate receptor 5, a gene that codes for a protein in the glutamate-gated ionic channel family), and *BDKRB2* (bradykinin receptor B2, a gene that encodes a receptor for bradykinin). The authors identified strong associations to individual disorders, such as *GHRH* with anxiety disorders, *PREB* with eating disorders, and *GRIK5* with bipolar disorder. Remarkably, a functional SNP, rs945032, located in the promoter region of *BDKRB2*, was associated with three disorders (panic disorder, substance abuse, and bipolar disorder).

Hartz and colleagues [65], focusing on the nicotinic receptor subunit genes associated with nicotine dependence (*CHRNA5/CHRNA3/CHRNB4* on chromosome 15q25 and *CHRNB3/CHRNA6* on chromosome 8p11) investigated the relationship between nicotine dependence and bipolar disorder. The authors reported two main findings: the first is that bipolar disorder does not influence and modify the association between nicotine dependence and nicotinic receptor subunit genes, and the second main finding is that variants in *CHRNB3/CHRNA6* are independently associated with bipolar disorder.

Mandelli and colleagues [69] examined the relationship of three genes: *IL1b* (interleukin 1 cytokine family), *NOS1AP* (cytosolic protein that interacts with the nNOS signaling molecule), and *TRPM2* (the protein encoded by this gene forms a tetrameric cation channel that is permeable to calcium, sodium, and potassium and is regulated by free intracellular ADP-ribose). The authors, thus, analyzed six polymorphisms (rs1143634, rs1143627, rs16944, rs1143623, rs12742393, and rs1556314) in association with BD and SUD comorbidity in a small but well clinically characterized sample of 131 BD patients and 64 healthy controls. Data support the involvement of *IL1b* in SUD but not in BD. The other genes investigated, *NOS1AP* and *TRPM2*, do not seem to play a crucial role in BD and BD comorbid for SUD.

Novak and colleagues [70] have performed an association analysis of single nucleotide polymorphisms (SNPs) within the *NR4A* genes with smoking behavior both in a cohort of 204 schizophrenia patients and 319 patients with bipolar disorder patients. The authors analyzed six polymorphisms (rs2603751, rs2701124, rs12803, rs834835, rs1131339, and rs1405209) of three genes that encode members of the steroid-thyroid hormone-retinoid receptor superfamily: *NR4A1*, *NR4A2*, and *NR4A3*. They showed that the *NR4A3* marker rs1131339 is significantly associated with the risk of smoking and the degree of smoking in a population of individuals with bipolar disorder. These results represented an essential confirmation of the involvement of the *NR4A3* gene in nicotine addiction in patients with mental health diseases.

Prossin and colleagues [71] tested whether the presence/absence of the *TSPO* functional polymorphism (rs6971) predicted differences in cortisol’s diurnal rhythm in healthy control and BD volunteers with and without comorbid AUD. The authors hypothesized the presence of interactions between diagnosis (BD, comorbid AUD), *TSPO* variant, and diurnal cortisol rhythm in study volunteers. In particular, the authors examined the rs6971 polymorphism of gene *TSPO*: the gene, present mainly in the mitochondrial compartment of peripheral tissues, encodes a protein that interacts with some benzodiazepines and has different affinities than its endogenous counterpart. The study concludes that, within Bipolar patients, the presence or absence of rs6971, the genetic variant of *TSPO*, may contribute to comorbidity of stress-exacerbated illnesses, including alcohol use disorders. Structural changes to the TSPO protein may be clinically impactful in humans with bipolar disorder and stress-exacerbated disease.

Sharp and colleagues [73] investigated the association of *TACR1* with bipolar disorder (BD), BD comorbid with alcohol dependence, alcohol dependence syndrome (ADS), and attention deficit hyperactivity disorder (ADHD). The authors analyzed six polymorphisms (rs3771833, rs3771829, rs3771856, rs17011370, rs1106854, and rs13387833) belonging to the *TACR1* gene. The gene belongs to a gene family of tachykinin receptors, and rs3771829 was associated with BD, ADS, and bipolar disorder comorbid with alcohol dependence compared with healthy controls. DNA sequencing in cases of BD and ADHD who had inherited *TACR1*-susceptibility haplotypes identified 19 SNPs in the promoter region, 50 UTR, exons, intron/exon junctions, and 30 UTR of *TACR1* that could increase vulnerability to BD, ADS, ADHD, and BP comorbid with ADS. Furthermore, rs1106854 was associated with BD, although a regulatory role for rs1106854 is unclear. The association with *TACR1*, BD, ADS, and ADHD suggests shared molecular pathophysiology between these disorders.

#### 3.4.5. Unspecific Polymorphisms Studies

Three studies did not investigate specific polymorphisms as the studies mentioned before did. However, those studies used three different methods for analyzing a large pool of genes.

Chang and colleagues [61] compared the frequencies of the *ADH1B* and *ALDH2* genotypes in two groups of Han Chinese in Taiwan: (1) bipolar disorder with and without alcohol disorder and (2) healthy controls. The authors analyzed six alleles of two genes involved in alcohol metabolism: *ALDH2* and *ADH1B*. The authors found that having bipolar disorder-II and alcohol disorder may be related to *ALDH2* and *ADH1B*, but having bipolar disorder-I and alcohol disorder may be related only to *ALDH2*.

Lydall and colleagues [67] aimed to identify genetic regions associated with BD comorbid with alcoholism and distinguish between genes involved in alcoholism and those involved in bipolar disorder. The authors conducted a genome-wide association study (GWAS) in which they genotyped 372 193 SNPs. The authors have found 3799 SNPs nominally associated with alcoholic bipolar patients. This genome-wide study replicated five previous associations with alcoholism and addiction that seems independent of BD. The most significant SNP associations with BP comorbid with Alcohol dependence syndrome were in or near genes involved in cell adhesion, neurotransmitter pathways, enzymatic activity, cellular messengers, connective tissue, and cell regulation. However, none of these reached genome-wide significance; therefore, the associations must be considered only as suggestive. Critical negative findings include the genes of the GABA system, but none of the GABA receptor genes, including *GABRA2*, at the corrected gene-wise level was identified. The authors concluded that this finding might reflect the small study size, genetic heterogeneity, or possibly different pathways mediating alcoholism in BD.

Reginsson and colleagues [72] hypothesized that the polygenic risk scores (PRSs) for schizophrenia (SCZ-) and bipolar disorder (BD) could be used to reveal the extent of genetic overlap between these psychotic disorders and substance use disorders. The authors used SCZ and BPD PRSs to predict smoking and addiction to nicotine, alcohol, or drugs in people who had not been diagnosed with psychotic disorders. PRSs from 144,609 subjects, including 10,036 people admitted to in-patient addiction treatment and 35,754 smokers, were used. They discovered that diagnoses of various substance use disorders and smoking were strongly associated with PRSs for SCZ (*p* = 5.3 1050–1.4 106) and BPD (*p* = 1.7 109–1.9 103), indicating a shared genetic etiology between psychosis and addiction. Those results supported the notion of common genetic roots of the observed comorbidity between addiction and psychotic disorders SCZ and BPD instead of solely being a direct consequence. The authors note that the risk of substance use disorders seems slightly less for the BPD-PRS than the SCZ-PRS.

## 4. Discussion

### Summary of Evidence

The current systematic review, which investigated the genetic polymorphisms associated with BD comorbid to SUD, discovered nine polymorphisms related to seven genes with significant relationships.: rs11600996 (*ARNTL*), rs228642/rs228682/rs2640909 (*PER3*), PONQ192R (*PON1*), rs945032 (*BDKRB2*), rs1131339 (*NR4A3*), rs6971 (*TSPO*), and rs1106854 (*TACR1*). We have also found three studies that did not analyze any specific polymorphism but used three different methods to analyze a large pool of genes.

There were no significant polymorphisms associated with BD + SUD concerning serotonergic and dopaminergic systems. Huang and colleagues found an association of the *h5-HTR1B* G861C locus with SUD but not BD. Mazza and colleagues, conversely, found that the G allele of the rs6318 polymorphism of the *5HTR2C* gene is associated with BD but not with BD + SUD. Both Gorwood and Szczepankiewicz underlie how the analyzed polymorphisms may not be involved in the shared genetic vulnerability of bipolar disorder comorbid to alcohol abuse.

Regarding the circadian rhythm regulation system, the two polymorphisms, rs11600996 (*ARNTL*) and rs228642 (*PER3*), are linked with the risk of BD comorbid to SUD in the male population. Benach and colleagues found that allele T is characteristic of a group of BD patients in SNP rs228642. The authors found no significant differences in genotype distribution and allele frequencies in female bipolar patients with comorbid AAD.

The authors also analyzed the group of BD patients with comorbid AAD and a family history of affective disorder. They observed a significant association of *PER3*: rs228682 TT and rs2640909 TT genotypes in patients with family history. Unfortunately, the results did not maintain statistical significance after applying the Bonferroni correction. In summary, those results supported the hypothesis that comorbidity of BD and Alcohol Abuse Disorder may be related to the polymorphisms of clock genes analyzed.

Bortolasci and colleagues analyzed PONQ192R polymorphism in two different studies. In the first one, the authors underlie that the odds of bipolar disorder were increased by the QQ genotype in smokers, suggesting that *PON1* Q192R polymorphisms mediate the effects of smoking on bipolar disorder. This interaction may be explained by the impact of functional *PON1* Q192R genotypes and smoking on immune-inflammatory and Oxidative and Nitrosative Stress (O&NS) pathways. A second significant finding of the first study is that smoking interacts with the functional *PON1* genotypes to predict bipolar disorder and decreases plasma *PON1* activity. The *PON1* lowering effects of smoking may contribute to inflammatory and oxidative burden, and thus, to the increased incidence of degenerative diseases. Likewise, the same pathways may partly account for the association between smoking and bipolar disorder. Bortolasci and colleagues’ second study sought to investigate the relationship between TRAP levels and *PON1* activity, *PON1* Q192R functional genotypes, smoking, interactions between *PON1* genotypes and smoking, and mood disorders, while controlling for ethnicity, marital status, body mass index (BMI), and gender. The findings of this study are that TRAP was significantly related to plasma *PON1* activity, *PON1* Q192R genotypes, an interaction between smoking and these genotypes, BMI, and gender. In addition, subjects with mood disorders displayed lowered TRAP levels than controls, although these effects were not significant in the final regression analysis. Tobacco Use Disorder (TUD) subjects showed lower levels of TRAP than controls, but these effects were no longer significant after considering the interaction between current smoking and *PON1* Q192R genotypes. TRAP levels were found to be significantly related to plasma *PON1* activity. PON1 is an HDL-associated enzyme with multifunctional activities, including antioxidant properties.

Gratacòs and colleagues studied the role of rs945032 (*BDKRB2*); this SNP is associated with three diagnoses, panic disorder, substance abuse, and bipolar disorder, remaining significant after correction in the case of panic disorder.

Novak and colleagues studied the rs1131339 polymorphism (*NR4A3*) and have found it is associated with the risk of smoking in subjects with BD. This study is the first to link SNPs in the *NR4A3* gene to drug addiction.

Prossin and colleagues underlie how, within Bipolar patients, the presence or absence of the common, functional TSPO polymorphism (rs6971) may contribute to comorbidity of stress-exacerbated illnesses, including alcohol use disorders.

Sharp and colleagues replicated a significant association with intron 1 *TACR1* mutations in BD in the BD with Alcohol dependence subgroup and Alcohol dependence syndrome cases compared to a screened population of healthy controls. Sequencing of *TACR1* in BD + AD and ADHD cases detected one novel base pair change in the 3′ UTR, although this was not significantly associated with BD compared to healthy controls. It was found that one marker, rs1106854, is positively associated with BD. The authors concluded that polymorphisms in *TACR1* increase susceptibility substantially to BPAD, ADS, and ADHD. The significant *TACR1* allele frequency difference between screened and controls suggested an effect of *TACR1* on normal drinking behavior.

## 5. Limitations

Several limitations require interpreting the results of this review with caution:The number of studies exploring associations between many genetic polymorphisms and BD comorbid to SUD is limited, and in some cases, the sample size used was small.The relationship between some polymorphisms and BD comorbid to SUD could also be influenced by gene–gene or gene–environment interactions.We could not pool data collected for a meta-analysis due to the heterogeneity of the genetic polymorphisms observed.

## 6. Conclusions

To the best of our knowledge, the current systematic review is the first to investigate the presence of genetic polymorphisms associated with BD comorbid to SUD. All in all, this systematic review lists potentially valuable genetic polymorphisms associated with BD and SUD.: rs11600996 (*ARNTL*), rs228642/rs228682/rs2640909 (*PER3*), PONQ192R (*PON1*), rs945032 (*BDKRB2*), rs1131339 (*NR4A3*), and rs6971 (*TSPO*). It is critical to mention that none of those findings have been confirmed by two or more studies. Therefore, all the polymorphisms indicated in this review need further evidence to be confirmed.

## Figures and Tables

**Figure 1 genes-13-01303-f001:**
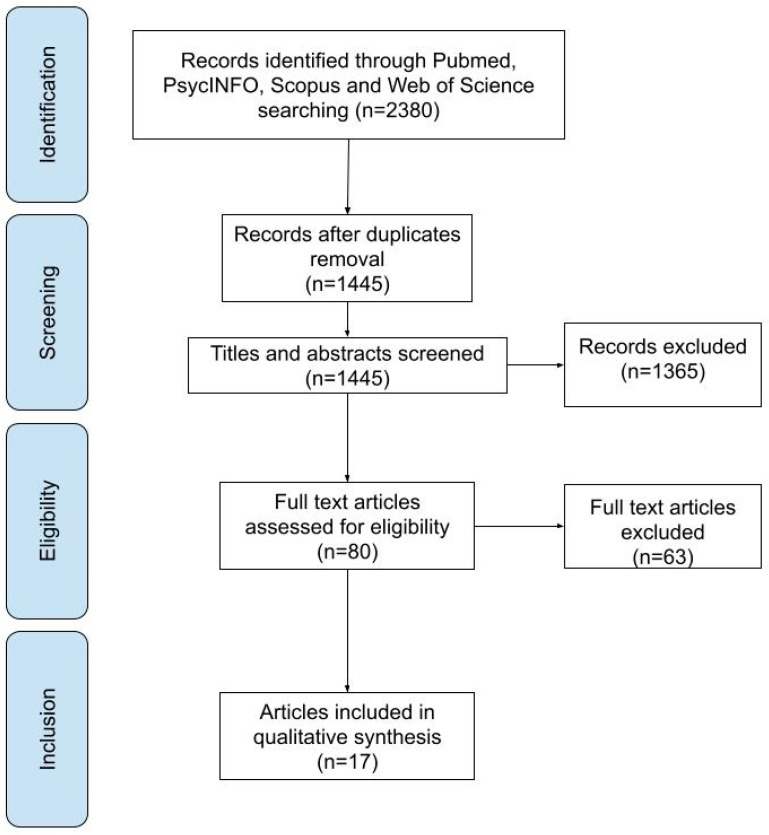
Flowchart of the search process.

**Table 1 genes-13-01303-t001:** The present table reports, in alphabetical order, the most relevant information of each eligible selected publication (NA = not available).

Author(s) and (Year)	Groups (N)	Diagnostic Criteria of BD/SUD	Gene(s)	Main Findings	NOS Scale
Banach et al. (2018) [58]	436 BD (76 AAD) M = 199 (47 ± 14), F = 237 (44 ± 15) 417 CTRs M = 202 (45 ± 8), F = 215 (46 ± 6)	DSM-IV	*CLOCK* (Clock Circadian Regulator) *ARNTL* (Aryl Hydrocarbon Receptor Nuclear Translocator Like) *TIMELESS* (Timeless Circadian Regulator) *PER3* (Period Circadian Regulator 3)	*ARNTL* gene (rs11600996) and *PER3* gene (rs228642) polymorphisms were associated with an increased risk of BD/AAD in a group of male patients; moreover, two other polymorphisms of the *PER3* gene, rs228682 and rs2640909, were associated with both AAD and family history of affective disorders.	8
Bortolasci et al. (2014) [59]	45 BD (Smoking 31) M = 12. F = 33 (44.5 ± 8.9) 91 MD (Smoking 36) M = 22, F = 69 (47.0 ± 8.2) 199 CTRs (Smoking 77) M = 82, F = 117 (46.4 ± 8.3)	DSM-IV	*PON1* (Paraoxonase 1)	Lowered plasma *PON1* activity was a trait marker of major depression and that *PONQ192R* gene–environment (such as smoking) interactions differentially predict the odds of depression and bipolar disorder.	7
Bortolasci et al. (2014a) [60]	45 BD 91 MD 197 CTRs 143 TUD (between all groups) M = 115, F = 218	DSM-IV	*PON1* (Paraoxonase 1)	TRAP levels were significantly associated with higher plasma *PON1* activity, the RR functional genotype, non-smoking by RR carriers, male gender and a higher BMI. TRAP levels were significantly lower in patients with mood disorders than in controls, but this association was no longer significant after considering the effects of the above predictors. The risk in the subgroup with low TRAP levels is increased by a smoking X RRenotype interaction and decreased by male gender, the RR genotype, and higher BMI and *PON1* activity. Plasma *PON1* activity, the *PON1* Q192R functional genotypes and specific interactions between this genotype and smoking contribute significantly to TRAP levels. Gender and BMI also appear to influence TRAP levels.	7
Chang et al. (2015) [61]	530 BP-I (59 AD) M = 277, F = 253 (34.32 ± 12.35) 788 BP-II (135 AD) M = 413, F = 375 (32.37 ± 11.53) 672 CTRs M = 442, F = 230 35.65 ± 10.68	DSM-IV-TR SADS-L	*ALDH2* (Aldehyde Dehydrogenase 2 Family Member) *ADH1B* (Alcohol Dehydrogenase 1B (Class I), Beta Polypeptide)	BP-II + AD associated to *ALDH2* and *ADH1B*, but BP-I + AD associated only to *ALDH2*.	7
Cui et al. (2011) [62]	182 PAFDs M = 73, F = 109 (NA) 214 CAFDs M = 101, F = 113 (39.67 ± 10.98) 472 CTRs M = 197, F = 275 (30.21 ± 11.04)	DSM-III-R and DSM-IV	*NGFB* (Nerve Growth Factor BETA subunit)	In women, *NGFB* gene was associated with primary affective disorders (PAFDs) but not with substance disorder comorbid with AFDs (CAFDs).	8
Gorwood et al. (2000) [63]	21 BD + AD M = 14, F = 7 (45.8 ± 9.45) 31 BD M = 15, F = 16 (53.4 ± 10.7) 35 AD M = 25 (46.2 ± 9.8) 35 CTRs M = 25 (43.5 ± 6.5)	DSM-III-R	*DRD2* (Dopamine Receptor D2)	In this sample, there is no evidence for a significant influence of the A1 allele of the D2 dopamine receptor gene in the specific association between bipolar disorder and alcohol dependence.	8
Gratacòs et al. (2008) [64]	165 substance abuse disorders M = 72.6% (39.0) 341 anxiety disorders, consisting of 173 panic disorder patients and 168 obsessive-compulsive patients M = 24.1% (36.02) M = 34.0% (35.9) M = 7.4% (24.4) 229 eating disorders M = 65.4% (45.6) 917 SCZ M = 44.3% (44.9) 625 affective disorders consisting of 203 bipolar disorders and 422 major depressions M = 32.7% (49.5) M = 51.7% (46.8) 2 CTRs groups: 607 CTRs 330 CTRs M = 56.1% (56.1)	DSM-IV	The comprehensive search set rendered 10 physiological pathways and a total of 338 candidate genes.	*GHRH* (growth hormone releasing hormone) was associated with anxiety disorders, *PREB* (prolactin regulatory element) was linked with eating disorders, and *GRIK5* (ionotropic kainate glutamate receptor 5) was correlated to bipolar disorder. Moreover, rs945032, a functional SNP of the bradykinin receptor B2 gene (*BDKRB2*) was associated to three disorders (panic disorder, substance abuse, and bipolar disorder).	8
Hartz et al. (2011) [65]	916 BD (367 Nicotine dependent) M = 359, F = 557 (43.5 ± 13.2) 1028 CRTs (269 Nicotine dependent) M = 599, F = 429 (52.9 ± 17.7)	DSM-IV	*CHRNB3* (Cholinergic Receptor Nicotinic Beta 3 Subunit) *CHRNA5* *CHRNA3* (Cholinergic Receptor Nicotinic ALFA 3 E 5 Subunit)	Bipolar disorder does not modify the association between nicotine dependence and nicotinic receptor subunit genes. Besides, variants in *CHRNB3/CHRNA6* are independently associated with bipolar disorder.	7
Huang et al. (2003) [66]	52 BD 208 MD 83 SCZ 96 CTRs M = 51, F = 45 (40.97 ± 15.9)	DSM-III-R	*HTR1B* (5-Hydroxytryptamine Receptor 1B)	There was an association between substance abuse disorder and major depression with the *h5-HTR1B* G861C locus in the patient population. This association was not found in bipolar disorder, schizophrenia, alcoholism, and suicide.	7
Lydall et al. (2011 [67]	506 BD NA 510 CTRs NA	ICD-10 RDC DSM-III-R	GWAS genome-wide association study	The authors conducted a genome-wide association study (GWAS) in which they genotyped 372 193 SNPs. The authors found 3799 SNPs nominally associated with alcoholic bipolar patients.	8
Mazza et al. (2010) [68]	131 BD (65 SUD) 65 CTRs F (*n* = 100), M (*n* = 96)	DSM-IV	*5HTR2C* (serotonin 2C receptor)	Evidence of the association between the functional rs6318 polymorphism of *5HTR2C* in bipolar disorder.	7
Mandelli et al. (2011) [69]	131 BD (66 SUD) 64 CTRs F = 49.7%	DSM-IV	*IL1b* (Interleukin 1 Beta) *NOS1AP* (Nitric Oxide Synthase 1 Adaptor Protein) *TRPM2* (Transient Receptor Potential Cation Channel Subfamily M Member 2)	This study does not support a role of *IL1b*, *NOS1AP*, and *TRPM2* in BD. *IL1b* may have a role in SUD.	7
Novak et al. (2010) [70]	319 BD (152 Smokers) M = 121. F = 198 (46.6) 204 Schizophrenia (126 Smokers) M = 139, F = 65 (43) 500 CTRs Matched for age, gender and ethnicity	DSM-IV ICD 10	*NR4A1* *NR4A2* *NR4A3* (Nuclear Receptor Subfamily 4 Group A Member 1, 2 E 3)	*NR4A3* marker rs1131339 is significantly associated with the risk of smoking, as well as with the degree of smoking, in a population of individuals with bipolar disorder.	8
Prossin et al. (2018) [71]	107 BD-I M = 32, F = 56 (41.9 ± 12.1) (50 AUD) M = 8, F = 18 (40.6 ± 16.6) 28 CTRs	DSM-IV	*TSPO* (Translocator Protein)	Interactions were confirmed both for *TSPO* with BD and *TSPO* with AUD	7
Reginsson et al. (2017) [72]	F: SCZ = 600 (35.7) BPD = 772 (63.3)	NA	NA	There are evidences of common genetic roots of the observed comorbidity between addiction and psychotic disorders SCZ and BPD.	7
Sharp et al. (2014) [73]	1099 BP (143 AD) NA 997 AD NA 1056 CTRs NA	RDC SADS-L OPCRIT DSM-IV ICD-10	*TACR1* (Tachykinin Receptor 1)	Authors report a replicated significant association with intron 1 *TACR1* mutations in BD in the BD + ADS subgroup and ADS cases in comparison with a population of healthy controls.	6
Szczepankiewicz et al. (2006) [74]	317 BD (42 AA) M = 131 (30 ± 11), F = 186 (33 ± 12) 350 CTRs M = 139 (41 ± 12), F = 211 (40 ± 11)	DSM-IV	*DRD1* *DRD2* *DRD3* *DRD4* (dopamine receptor D1, D2, D3, D4)	The analyzed polymorphisms of the dopamine genes may not be associated in the shared genetic vulnerability to bipolar disorder and alcohol abuse.	8

## Data Availability

Data are available upon request from the corresponding author.

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
