# Peer review of "A Systematic Review of Genetic Polymorphisms Associated with Bipolar Disorder Comorbid to Substance Abuse"

_genes, 2022, doi:10.3390/genes13081303_

Round 1
Reviewer 1 Report
My suggestions:
1. In the introduction, instead of "Thanks to Genome-wide Association Studies (GWAS), the understanding of affective disorders has significantly improved" I would add something else. For example, "In recent years, due to GWAS technologies, the knowledge about disease risk genes has been improved"
2. Also in the introduction, what is the function of, DCLK3? Please, mention it briefly.
3. Was ALDH2 studied only in East Asia and Thailand? How about in Europe or other populations?
4. Table 1 seems a little bulky, but I understand, since it involves a lot of information. However, better to use single spacing at the table. You may add which populations have been examined in the studies, included in the table. Also, please add the table history.
5. A few figures would definitely improve this manuscript. It would be nice to see some pathway figure on Serotonergic Genes, dopaminergic Genes, and other genes, and how they may result in bipolar disease.
Author Response
Dear Reviewer 1,
Thanks for your important suggestions. Here we report all the modifications that we did base on your tips.
1) We accepted your suggestion relative to the sentence of the introduction; Thanks to Genome-wide Association Studies (GWAS), the understanding of affective disorders has significantly improved; we changed it with the phrase prompted by you (lines 48-50).
2) Regarding the second point, we didn't explain much about the DCLK3 gene. Now there are reported several gene functions.
3) The most relevant evidence in the literature about ALDH2 is about East-Asian subjects. We have reported some evidence about the genetic association between alcohol consumption and western people: this evidence suggests a crucial role for ADH1B [45]. Thanks to your question, we now think there is a better vision for that.
4) Table 1 has been modified. We operated a graphical and content revision to reduce its size. We added a brief table history.
5) Since we deal with serotonergic and dopaminergic genes in the results (because of some eligible selected publications), where no significant genetic association with BD comorbid to SUD where reported, we believe
that those gene's pathway figures on BD could be redundant.
Reviewer 2 Report
This review aims to provide a set of polymorphisms shared between BD and SUD comorbidities such as licit and illicit drug use. The review is well organized. I have a few comments only:
1) One general comment is was not clear from reading the manuscript, whether the authors accounted for LD patterns, i.e., did they look whether the same SNP is associated with BD and SUD, or allowed for other SNPs in high LD with, let say, the risk SNP of BD, to be also considered?
2) I applaud the authors for considering potential bias in evaluating the original research manuscripts. I was not clear whether they included in the final analysis only studies having a Newcastle-Ottawa score greater than 7.
3) I could not discern how the initial screening was performed, whether the authors screened all 2380 research manuscripts by reviewing the entire manuscript or just pursuing the abstracts.
4) it would be helpful if the authors also commented on the analytical approaches and how well these were justified in the original study to confirm (or reject) the association of the polymorphisms with both BD and SUD.
Author Response
Dear Reviewer 2
Thanks for your important suggestions. Here we report all the modifications that we did base on your tips.
1) We did not account for LD patterns. We thought this review did not aim to analyze whether an SNP is associated with BD and SUD by Linkage Disequilibrium Patterns.
2) First, we screened and included the eligible selected publications; then, we calculated each publication's NOS score. None of them had a score lower than 6.
3) We specify in lines 141-143 that in the first screening, the evaluators read only titles and abstracts to select papers that met the inclusion criteria.
4) The analytical approaches of the original studies are quite different. To this aim, based on the potential bias information provided by the NOS scale, we think that all the studies used the correct approach. Nonetheless, it is essential to note that none of the reported findings have been confirmed by two or more studies; thus, we believe that all the polymorphisms identified in this review require additional evidence to be confirmed.